# FedProp: Cross-client Label Propagation for Federated Semi-supervised Learning

## Abstract

Federated learning (FL) allows multiple clients to jointly train a machine learning model in such a way that no client has to share their data with any other participating party. In the supervised setting, where all client data is fully labeled, FL has been widely adopted for learning tasks that require data privacy. However, it is an ongoing research question how to best perform federated learning in a semi-supervised setting, where the clients possess data that is only partially labeled or even completely unlabeled. In this work, we propose a new method, *FedProp*, that follows a manifold-based approach to semi-supervised learning (SSL). It estimates the data manifold jointly from the data of multiple clients and computes pseudo-labels using cross-client label propagation. To avoid that clients have to share their data with anyone, *FedProp* employs two cryptographically secure yet highly efficient protocols: *secure Hamming distance computation* and *secure summation*. Experiments on three standard benchmarks show that *FedProp* achieves higher classification accuracy than previous federated SSL methods. Furthermore, as a pseudo-label-based technique, *FedProp* is complementary to other federated SSL approaches, in particular consistency-based ones. We demonstrate experimentally that further accuracy gains are possible by combining both.

## 1 Introduction

Federated Learning (FL) is a machine learning paradigm in which multiple clients, each holding their own data, cooperate to jointly train a model. Training is coordinated by a central server, which, however, must not have direct access to client data. Typically this is not due to the server being viewed as a hostile party but rather to comply with external privacy and legal constraints that require client data to remain stored on-device. FL has been receiving abundant interest in recent years as it allows models to be trained on valuable data that would otherwise be inaccessible. To date, the vast majority of research within FL has been focused on the supervised setting, in which client data is fully labeled. However, in many real-world settings, this is not the case. For instance, in *cross-device* FL, smartphone users are not likely to be interested in annotating more than a handful of the photos on their devices or in a *cross-silo* setting the labeling of medical imaging data may be both costly and time consuming. As such, in recent years there has been growing interest in learning from partly labeled data in a federated setting.

In this work we propose *FedProp*, a method for semi-supervised learning (SSL) in the federated setting that follows a manifold-based approach to pseudo-labeling client data. During each training round *FedProp* leverages the data of multiple clients to obtain an estimate of the data manifold which it then uses to compute pseudo-labels for clients' unlabeled data via label propagation. Using these pseudo-labels clients then train in a supervised manner for the remainder of the round. The motivation for this approach comes from the fact that the more data that is available, the more densely we have sampled the manifold and therefore the better our estimates and pseudo-labels will be. Thus, it is of crucial importance to be able to combine information from multiple clients, rather than treating each client's data in isolation. The key challenge lies in how to perform such cross-client pseudo-labeling, the steps of which would normally require the data of participating parties to be shared in order to estimate the manifold and compute label propagation.

Our main contribution lies in the `CrossClientLP` subroutine of *FedProp*, which we propose to address this challenge. It uses locality-sensitive hashing and secure Hamming distance computa-

tion to efficiently estimate the cross-client data manifold. It then distributes the label propagation computation across clients and aggregates the results using *secure summation*. `CrossClientLP` preserves privacy, in the sense that it does not require clients to share their data with anyone else. At the same time, it adds only limited communication and computation overhead relative to popular federated learning methods such as *FederatedAveraging*.

Our experiments show that *FedProp* outperforms all existing methods for federated semi-supervised learning, as well as a range of natural baselines in the standard CIFAR-10 setup. Going beyond prior work, we also evaluate *FedProp* on more challenging datasets, namely CIFAR-100 and Mini-ImageNet, where we also observe substantial improvements in accuracy. Moreover, as a method for pseudo-labeling unlabeled data, *FedProp* is orthogonal to other approaches for federated SSL, in particular those based on consistency regularization. We demonstrate this empirically by combining *FedProp* with such an approach and observing that this often leads to further accuracy gains.

## 2 RELATED WORK

**Semi-supervised Learning**    Semi-supervised learning (SSL) is a classical and well studied problem in machine learning where the goal is to leverage both labeled and unlabeled training examples to improve performance on some task, see (Chapelle et al., 2006) for a full overview. In recent years there has been a great deal of interest in applying deep learning techniques to SSL. Broadly speaking such semi-supervised deep learning approaches can be categorized into two groups. The first group consists of methods that add an unsupervised loss term to the objective function. In particular many of these methods introduce some form of consistency regularization (Sajjadi et al., 2016), which encourages the model to produce similar outputs for similar inputs, examples include (Tarvainen & Valpola, 2017; Berthelot et al., 2019; Xie et al., 2020). The second group consists of methods that exploit unlabeled data by computing pseudo-labels for unlabeled points and then training on these in a supervised fashion, for instance (Lee, 2013; Shi et al., 2018; Iscen et al., 2019; Rizve et al., 2021). Combinations of both approaches are also possible (Iscen et al., 2019; Sohn et al., 2020).

**Federated Learning**    Federated learning (FL) (McMahan et al., 2017) was originally proposed for learning on private fully labeled data split across multiple clients. For a survey on recent developments in the field see (Kairouz et al., 2021). A number of recent works propose federated learning in the absence of fully labeled data. When only unlabeled data is available, methods for cluster analysis, dimensionality reduction have been proposed (Dennis et al., 2021; Grammenos et al., 2020). Federated self-supervised learning (Zhuang et al., 2022; Makhija et al., 2022) can also be performed where the goal is to learn representations of unlabeled data that can later be fine-tuned to other tasks. However, all of these settings are different from the task of semi-supervised learning, in which the goal is to directly learn better classifiers from labeled and unlabeled client data.

For semi-supervised FL, several works follow a consistency-based approach. Jeong et al. (2021) propose inter-client consistency and parameter decomposition to separately learn from labeled and unlabeled data. Long et al. (2020) apply consistency locally through client based teacher models. Zhang et al. (2021) and Diao et al. (2022) focus on an alternative setting in which the server has access to labeled data. In this setting Zhang et al. (2021) combine local consistency with grouping of client updates to reduce gradient diversity while Diao et al. (2022) combine consistency, through strong data augmentation, with pseudo-labeling unlabeled client data. Other methods focus exclusively on pseudo-labeling: Albaseer et al. (2020) and Lin et al. (2021) both use network predictions to assign pseudo labels, while Presotto et al. (2022) develop a specialized method for human activity recognition which uses label propagation locally on each client to pseudo label incoming data. Unlike in classical SSL, none of the above methods fully make use of the knowledge gained from estimating the data manifold because they exploit interactions between data points at most locally within each client. In contrast, *FedProp* uses securely computed cross-client interactions, thereby obtaining a better estimate of the data manifold.

## 3 PRELIMINARIES AND BACKGROUND

We assume a federated classification setting with $m$ clients coordinated by a central server. Each client $j$ possesses partly labeled data $(X_L^{(j)}, y_L^{(j)}, X_U^{(j)})$ with $X_L^{(j)} := \{x_1^{(j)}, \ldots, x_{l^{(j)}}^{(j)}\}$, $y_L^{(j)} :=$

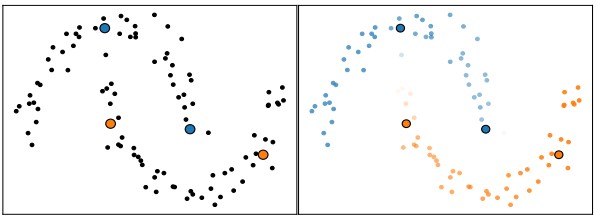 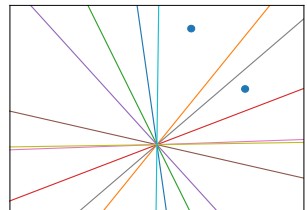

(a) *Label propagation*. Left: a dataset with 2 labeled data points per class (colored) and many unlabeled ones (black). Right: soft labels for the unlabeled data points are estimated by propagating label information along a 10-nearest-neighbor graph.

(b) *Locality sensitive hashing*. The expected number of random hyperplanes that separate two data points (here: 2 out of 10) is proportional to the angle between the points.

Figure 1: Illustration of *label propagation* (a) and *locality sensitive hashing* (b).

---

**Algorithm 1:** `LabelPropagation`

---

**Input:** data matrix $V$, matrix of partial labels $Y$

1   $A \leftarrow$ matrix of pairwise cosine similarities $A_{ij} = \frac{\langle v_i, v_j \rangle}{\|v_i\| \|v_j\|}$

2   $B \leftarrow$ sparsify $A$ to $k$-NN graph by $B_{ij} = \begin{cases} A_{ij} & \text{if } A_{ij} \in \text{top}_{k+1}\left((A_{is})_{s \in [n]}\right) \text{ and } i \neq j, \\ 0 & \text{otherwise} \end{cases}$

3   $W \leftarrow B + B^\top$

4   $\mathcal{W} \leftarrow D^{-\frac{1}{2}} W D^{-\frac{1}{2}}$ for $D = \text{diag}(W \mathbf{1}_n)$

5   $Z \leftarrow (\text{Id}_n - \alpha \mathcal{W})^{-1} Y$

6   $\hat{Z} \leftarrow$ normalize $Z$ via $\hat{Z}_{ij} = Z_{ij} / (\sum_{c=1}^{C} Z_{ic})$

7   $\hat{y} \leftarrow$ compute labels via $\hat{y}_i = \arg \max_{c \in [C]} \hat{Z}_{ic}$

8   $\omega \leftarrow$ compute weights via $\omega_i = 1 - H(\hat{Z}_i) / \log C$ with $H(p) = -\sum_{j \in [C]} p_j \log p_j$.

**Output:** pseudo-labels $\hat{y}$, weights $\omega$

---

$\{y_1^{(j)}, \ldots, y_{l^{(j)}}^{(j)}\}$ and $X_U^{(j)} \coloneqq \{x_{l^{(j)}+1}^{(j)}, \ldots, x_{n^{(j)}}^{(j)}\}$. Here $l^{(j)}$ and $n^{(j)}$ denote the number of labels and the total number of data points owned by client $j$. Let $D$ denote the data dimension, so that $x_i^{(j)} \in \mathbb{R}^D$, and $C$ the number of classes, so that $y_i^{(j)} \in [C] \coloneqq \{1, \ldots, C\}$. We further denote by $f_\theta : \mathbb{R}^D \to \mathbb{R}^C$ a parametric classifier of the data, with parameters $\theta$, which in our case is a deep neural network. We decompose $f_\theta$ into a feature extractor $f_\phi : \mathbb{R}^D \to \mathbb{R}^d$ and a classifier head $f_\psi : \mathbb{R}^d \to \mathbb{R}^C$, so that $f_\theta = f_\psi \circ f_\phi$ with $\theta = (\psi, \phi)$. In standard fashion, $f_\theta$ will be trained by stochastic gradient optimization of a loss function $L(X, Y, \omega; \theta) = \sum_{i=1}^{n} \omega_i \ell(f_\theta(x_i), y_i)$, where $\omega_i$ are per-sample weights and $\ell$ is the cross entropy loss.

Our aim is to learn the classifier $f_\theta$ by making use of all client data, both labeled and unlabeled, while respecting the key principle of a federated learning setting, namely that clients should not be required to share their data with anyone else. To this end, we build on two established techniques: *label propagation* and *locality-sensitive hashing*, which we now introduce.

### 3.1   Label Propagation (LP)

Given partly labeled data, LP is a classic technique for assigning (pseudo-)labels to the unlabeled data. We describe a variant of the graph-based diffusion approach of Zhou et al. (2004). The key idea is to propagate label information from labeled to unlabeled points taking into account the geometry of the underlying data manifold, see Figure 1a for an illustration. The manifold is approximated by a neighborhood graph of the data, in which points are connected based on how similar they are.

Algorithm 1 describes the procedure in pseudocode. As input, it takes a set of data vectors, $V = V_L \cup V_U = \{v_1, \ldots v_l, v_{l+1}, \ldots v_n\}$, and partial labels $\{y_1, \ldots, y_l\}$ from $C$ classes, which we encode in one-hot matrix form: $Y \in \mathbb{R}^{n \times C}$ with $Y_{ic} = \mathbb{1}_{(y_i = c)}$ for $i \leq l$ and $Y_{ic} = 0$ otherwise. The algorithm starts by computing a matrix $A \in \mathbb{R}^{n \times n}$ of pairwise similarities between all data

---

**Algorithm 2:** FedProp

---

**Input:** labeled data $(X_L^{(j)}, y_L^{(j)})_{j=1}^m$, unlabeled data $(X_U^{(j)})_{j=1}^m$      *// stored on-device at clients*

1   $\theta \leftarrow$ initialize randomly

2   $\theta \leftarrow \texttt{FederatedOptimization}(X_L^{(j)}, y_L^{(j)})$      *// initial training on labeled examples*

3   **for** *round* $t \in [1, \dots T]$ **do**

4      $P \leftarrow$ server randomly selects $\tau m$ clients

5      Server broadcasts $\theta$ to each client in $P$

6      **for** *client* $j \in P$ *in parallel* **do**

7          $V^{(j)} \leftarrow f_\phi(X^{(j)})$ with $X^{(j)} := X_L^{(j)} \cup X_U^{(j)}$    *// embed labeled and unlabeled data*

8          $\hat{y}^{(j)}, \omega^{(j)} \leftarrow \texttt{CrossClientLP}(V^{(j)}, Y^{(j)}, P, \text{Server})$    *// compute pseudo-labels*

9          $\theta^{(j)} \leftarrow \texttt{ClientUpdate}(X^{(j)}, \hat{y}^{(j)}, \omega^{(j)}, E; \theta)$    *// train with pseudo-labels*

10         Client $j$ sends $\theta^{(j)}$ to the server

11      $\theta \leftarrow \texttt{ServerUpdate}\big((\theta^{(j)})_{j \in P}\big)$

**Output:** model parameters $\theta$

---

points (line 1). $A$ is sparsified by keeping only each point's $k$ nearest neighbors, symmetrized, and then normalized (lines 2–4), resulting in a transition matrix, $\mathcal{W}$. Label propagation now amounts to repeatedly spreading label information across the resulting graph by multiplying with $\alpha\mathcal{W}$ until convergence, where $\alpha \in (0, 1)$ is a hyper-parameter reflecting how much the algorithm should weight the original labels. The outcome of this can be computed in closed form (line 5). The resulting matrix, $Z \in \mathbb{R}^{n \times C}$, can be interpreted as unnormalized class scores for each data point. Normalizing $Z$ across rows yields, for each point, a probability distribution across all classes (line 6). From this, one assigns a maximum-likelihood label to each point (line 7) as well as a weight reflecting the confidence in the assignment based on the entropy of class probabilities (line 8).

### 3.2 Locality-Sensitive Hashing (LSH)

LSH (Indyk & Motwani, 1998) is a procedure for hashing real-valued vectors into binary vectors while preserving their similarity. Let $v \in \mathbb{R}^d$ be a vector. To encode $v$ into a binary vector $b$ of length $L$, LSH randomly samples $L$ hyperplanes in $\mathbb{R}^d$. For each hyperplane it checks whether $v$ lies above or below it and sets the $i$th bit in $b$ as 1 or 0 accordingly. Formally, $b_i = \mathbb{1}_{\langle v, u_i \rangle \geq 0}$, where $u_i \in \mathbb{R}^d$ is the normal vector of the $i$th hyperplane. A key property of LSH is that it approximately preserves cosine-similarity. Concretely, for vectors $v_1, v_2$ with LSH encodings $b_1, b_2$, one has $\frac{\langle v_1, v_2 \rangle}{\|v_1\| \|v_2\|} \approx \cos(\pi H(b_1, b_2)/L)$ where $H$ is the Hamming distance (number of bits that differ) between two binary vectors. The reason is that the probability of $b_1$ and $b_2$ differing at any bit $i$ is the probability that the $i$th sampled hyperplane lies between $v_1$ and $v_2$, which is equal to $\angle(v_1, v_2)/\pi$, see Figure 1b. By the law of large numbers, the more hyperplanes one samples, the better the approximation quality.

## 4 FedProp: Label Propagation for Federated SSL

In the following section we describe our approach to the problem of federated semi-supervised learning. *FedProp*, shown in pseudocode in Algorithm 2, follows a general FL template of alternating local and global model updates. As such, it is compatible with existing FL optimization schemes such as *FederatedAveraging* (McMahan et al., 2017), *FedProx* (Li et al., 2020), or *SCAFFOLD* (Karimireddy et al., 2020). The choice of scheme, which we refer to as `FederatedOptimization`, determines the exact form of `ClientUpdate` and `ServerUpdate`.

The first stage of *FedProp* (lines 1–2) is to initialize the model with federated training on only the labeled examples. The second stage takes place over $T$ rounds. To start each round the server samples some fraction $\tau$ of the clients. Each sampled client gets the current model parameters from the server and embeds its labeled and unlabeled data with the feature extractor $f_\phi$. Clients and server then collaboratively execute a cross-client label propagation step on these embeddings (line 8), which we discuss in detail in Section 4.1. As output of this step each client gets pseudo-labels and weights for their unlabeled data, which they then use to run $E$ epochs of local supervised training. Finally, clients send the updated local models to the server which aggregates them (lines 10–11).

---

**Algorithm 3:** `CrossClientLP`

---

**Input:** set of clients $P$, client data $(V^{(j)}, Y^{(j)})_{j \in P}$      *// data stored on-device at clients*

1-XS:   Setup: clients exchange private and public keys, agree on random seeds

2-CS:   **for** *client $j \in P$ in parallel* **do**

3-CS:      $\Pi \leftarrow L$ random projections          *// same $\Pi$ for each client*

4-CS:      $B^{(j)} \leftarrow \text{LSH}(V^{(j)}, \Pi)$

5-XS:   $H \leftarrow \texttt{SecureHamming}((B^{(j)})_{j \in P})$      *// server gets Hamming matrix*

6-SS:   $A \leftarrow$ compute cosine similarity matrix from $H$      *// line 1 of Alg. 1*

7-SS:   $B \leftarrow$ sparsify $A$ to $k$-NN graph      *// line 2 of Alg. 1*

8-SS:   $W = B + B^\top$      *// line 3 of Alg. 1*

9-SS:   $\mathcal{W} \leftarrow D^{-\frac{1}{2}} W D^{-\frac{1}{2}}$      *// line 4 of Alg. 1*

10-SS:   $S \leftarrow (\text{Id}_n - \alpha \mathcal{W})^{-1}$      *// line 5 of Alg. 1 (server part)*

11-CS:   **for** *client $j \in P$ in parallel* **do**

12-CS:      $S_L^{(j)} \leftarrow \text{labeled-cols}_j(S)$      *// client gets columns corresponding to labeled data*

13-CS:      $\bar{Z}^{(j)} \leftarrow S_L^{(j)} Y_L^{(j)}$      *// line 5 of Alg. 1 (per-client part)*

14-XS:      $Z^{(j)} \leftarrow \texttt{SecureSum}\big((\text{rows}_j(\bar{Z}^{(k)})_{k \in P}\big)$      *// line 5 of Alg. 1 (cross-client part)*

15-CS:      $\hat{Z}^{(j)} \leftarrow \text{normalize}(Z^{(j)})$      *// line 6 of Alg. 1*

16-CS:      $\hat{y}^{(j)}, \omega^{(j)} \leftarrow$ pseudo-labels and weights      *// lines 7 − 8 of Alg. 1*

**Output:** pseudo-labels and weights $(\hat{y}^{(j)}, \omega^{(j)})_{j \in P}$      *// available only to respective clients*

---

## 4.1 CROSS-CLIENT LABEL PROPAGATION

To compute pseudo-labels, we would like to execute label propagation over a cross-client neighborhood graph built from the embedded data of all currently selected clients. The challenge lies in how to carry out this computation. By default, Algorithm 1 would require the clients to share the feature embeddings of their data in order to compute the similarity matrix $A$ (line 1) and to share their labels when computing the class score matrix $Z$ (line 5), thereby violating the constraints of the federated setting. A naive solution to overcoming this problem would be to execute Algorithm 1 in its entirety using *multi-party computation* (MPC) (Yao, 1982; Cramer et al., 2015), a set of cryptographic tools that—in principle—allow participants to jointly compute any function in a way that leaks no information to any of them. However, for complicated functions of real-valued inputs, such as matrix inversion, MPC has high computational and communication overhead and would not be practical.

Our main contribution lies in the `CrossClientLP` routine, which allows secure yet efficient computation of pseudo-labels based on cross-client label propagation. The key idea is to take a hybrid approach and divide the steps of the LP algorithm into three groups: steps that clients execute locally using only their own data (*client steps, CS*), steps that the server executes on aggregated data (*server steps, SS*), and steps that require computing quantities based on cross-client data (*cross steps, XS*).

Adopting the viewpoint in which the server is non-hostile, we trust the server to execute its steps on non-revealing aggregate data and return correct results of computations.[1] Client steps are also unproblematic from a privacy viewpoint, so what remains is how to execute the cross steps efficiently yet without leaking client data. To do so, we transform the problem into a form that is equivalent but allows for the application of tailored cryptographic tools, that are highly efficient yet fully secure.

Algorithm 3 shows pseudo-code for `CrossClientLP`. For simplicity of exposition, we describe it as if communication between clients were possible, which is a reasonable assumption in a *cross-silo* setting. If this is not possible, as is likely the case in *cross-device* FL, then communication is orchestrated by the server, see e.g. Bonawitz et al. (2017). First, the clients use a secure key exchange procedure to agree on a shared random seed that remains unknown to the server (line 1). This is a common step in federated learning when cryptographic methods are meant to be employed, see e.g. the description in Bonawitz et al. (2017). The clients use the agreed-on random seed to generate a common set of $L$ random projections (line 3). Each client then uses LSH to hash its

---

[1] In particular we do not consider malicious servers in the cryptographic sense, which would be allowed to employ attacks such as model poisoning or generating fake clients in order to break the protocol.

feature vectors into a binary representation (line 4), as this is the most efficient input format for cryptographic computation. The key insight here is that all clients should use the same random projections. That way, it is possible to recover the cosine similarity between any two feature vectors, even across clients, from the Hamming distance between their binary representations. To securely compute all Hamming distances, each pair of clients run a suitable cryptographic routine, such as SHADE (Bringer et al., 2013). From this, the server receives a matrix, $H \in \mathbb{N}^{n \times n}$, for $n = \sum_{j \in P} n^{(j)}$, containing the Hamming distances between all pairs of embedded feature vectors across all currently selected clients (line 5). Crucially, no clients need to share their data to make this possible and the security guarantees of the cryptographic protocols mean that no participating party (server or client) can learn anything from intermediate stages of the computation. With $H$ the server can now execute a number of steps by itself (lines 6–10): it converts $H$ to an estimate of the cosine similarity matrix $A$ using the relation from Section 3.2, it computes the normalized and symmetrized nearest-neighbors matrix, and then inverts that to obtain the propagation matrix, $S \in \mathbb{R}^{n \times n}$.

The next step of label propagation would be to compute $Z = SY$, where the matrix $Y \in \mathbb{R}^{n \times C}$ contains the label information of all participating clients. To do this without the clients having to share their labels, we express the computation as $Z = \sum_{j \in P} S^{(j)} Y^{(j)}$ where $S^{(j)} \in \mathbb{R}^{n \times n^{(j)}}$ is the sub-matrix of $S$ consisting of only the columns that correspond to the data of client $j$. In fact we can refine this further by observing that by definition the rows of $Y^{(j)}$ that correspond to the unlabeled data of client $j$ are identically 0 and hence do not contribute to the multiplication. We therefore let $Y_L^{(j)} \in \mathbb{R}^{l^{(j)} \times C}$ be the rows of $Y^{(j)}$ corresponding to labeled points and $S_L^{(j)} \in \mathbb{R}^{n \times l^{(j)}}$ be the corresponding columns $S^{(j)}$ and we still have that $Z = \sum_{j \in P} S_L^{(j)} Y_L^{(j)}$. This observation has allowed us to distribute the computation of $Z$ among the clients while minimizing the size of the matrices that must be transmitted. Therefore, each client $j \in P$ receives $S_L^{(j)}$ from the server (line 12) and locally computes $\bar{Z}^{(j)} = S_L^{(j)} Y_L^{(j)} \in \mathbb{R}^{n \times C}$, (line 13), which reflects the influence of $j$'s labels on all other data points. By now we have computed $Z$, but the result is additively split across the clients since $Z = \sum_{j \in P} \bar{Z}^{(j)}$. Each client $k$ requires only the rows of $Z$ that correspond to their own data points which we denote by $Z^{(k)} \in \mathbb{R}^{n^{(k)} \times C}$. This is computed by client $k$ using a secure summation routine over these rows of the matrices held by all other clients in $P$. Note that secure summation is commonly used in FL for model averaging (Bonawitz et al., 2017). Finally, each client $k$ locally computes their pseudo-labels and weights from $Z^{(k)}$ (lines $15 - 16$).

## 4.2 Analysis

In this section, we analyze the *correctness*, *privacy*, *efficiency* and *robustness* of Algorithm 3.

**Correctness**    Algorithm 3 implements the same computation as Algorithm 1, except it estimates cosine similarity via the Hamming distance of the LSH binary vectors. Thus, the output of Algorithm 3 approximates that of Algorithm 1. The approximation quality depends on $L$, the LSH vector length. In practice, we observe no difference in behavior already for small values, e.g. $L = 4096$.

**Privacy**    The main insight is that Algorithm 3 adheres to the federated learning principle that clients do not have to share their data or labels with any other party. This is ensured by the fact that all cross-client operations are computed using cryptographically secure methods. The only information seen by the server about client data is contained in the matrix of Hamming distances $H$, which approximates the matrix $W$ of cosine similarities between client feature vectors. While certainly influenced by the client data, we consider $W$ (and therefore $H$) a rather benign object for a non-hostile server to have access to: first, the similarity is not computed between input data itself, but its feature representation according to the current model. Second, even the feature vectors could not be reconstructed from $W$ because cosine similarity depends only on angles, so any rescaling and rotation of the feature vectors would result in the same $W$ matrix. Clients do not see $W$ but they see some of the columns of $S$. These reflect how their labeled data can influence all other datapoints according to the data manifold as estimated from the participating clients' feature vectors. However, this influence is unnormalized and hence the influence relative to other clients cannot be known.

**Efficiency – Computational**    Ordinarily, LP imposes only a small overhead compared to the computational cost of training the deep network on the clients. That is because the matrices it makes use of scale with the number of datapoints, rather than the (much larger) number of network

parameters. The same holds for `CrossClientLP`, which—with the exception of the additional LSH steps—performs the same computational steps as ordinary LP. Furthermore, the most expensive steps of normalization and matrix inversion are carried out by the server. LSH itself requires only one additional matrix multiplication per client which adds negligible overhead on top of the computation of client features. By design all cryptographic primitives are chosen to allow maximal efficiency. For the `SecureHamming` step, special-purpose routines, such as SHADE (Bringer et al., 2013), rely on *oblivious transfer (OT)* (Ishai et al., 2003; Roy, 2022) as the only cryptographic primitive. Highly optimized libraries exist for this, such as libOTe (Peter Rindal), which allows for hundreds of thousands of OT operations per second. `SecureSum` can be implemented efficiently using, e.g., *secret sharing* (Shamir, 1979) or *garbled circuits* (Yao, 1986). A similar secure summation step is already used in FL to avoid information leakage during model averaging (Bonawitz et al., 2017). A number of libraries exist that are tailored to this task, for instance SAFELearn (Fereidooni et al., 2021), which allows for aggregation of tens of megabytes per second. The computational overhead added by both of these methods is typically small compared to the training of the network itself.

**Efficiency – Communcation** `CrossClientLP` incurs additional communication costs at two steps of Algorithm 3. Let $p = |P|$ be the number of participating clients and $n = \sum_{j \in P} n^{(j)}$ the total number of data points they possess. For computing the Hamming matrix, each client $k$ has to communicate (in encrypted form) $(p-1)Ln^{(k)}$ bits to other clients, and the server receives $n^{(k)}n$ values. To propagate the labels via the distributed matrix multiplication, each client $k$ first receives from the server a matrix of size $n \times l^{(k)}$ and then transmits a matrix of size $n^{(j)} \times C$ to every other client $j \in P\backslash\{k\}$. For example, in our CIFAR-10 experiments, in total this adds up to approximately 2.5MB of data per client, whereas transferring the model to the client and back requires 24MB of payload per client. Thus, in this setting, *FedProp* adds an overhead of about $10\%$ over pure *FederatedAveraging*. When using larger networks the cost of sending model updates becomes even more dominant, e.g. in our MiniImageNet experiments the overhead drops to approximately $4\%$.

**Robustness** In *cross-device* FL clients may be unreliable and prone to disconnecting during training. Therefore, it is important that FL algorithms can still execute and make progress even in the event of client dropouts. This is indeed the case for *FedProp*: a client dropping out before the `SecureHamming` step (Algorithm 3, line 5), is equivalent to it not having been sampled by the server in the first place: its data will not be used for this round of label propagation and the subsequent model update. Because the Hamming computation is executed pairwise, a client dropping out during this step has no effect on the computation of other clients. The result will be missing entries in the Hamming matrix, $H$, which the server can remove, thereby leading to the same outcome as if the client had dropped out earlier. If clients drop out after $H$ has been computed, but before the `SecureSum` step (Algorithm 3, line 14), they will have contributed to the estimate of the data manifold, but they will not contribute label information to the propagation step. This has the same effect as if a client only had unlabeled data. The exact effect of dropouts during `SecureSum` depends on the characteristics of its underlying cryptographic implementation. For example, when using $(k, n)$-*threshold secret sharing*, the computation can be completed as long as at least $k$ out of $n$ clients participate until the end (Shamir, 1979). Any later dropout will only result in that client not contributing to the following model update step, but it will not affect the other clients' computations.

## 5 EXPERIMENTS

In this section, we evaluate the accuracy of *FedProp* against other methods for semi-supervised FL as well as report on ablation studies. As our emphasis here is on accuracy, not real-world efficiency, we use a simulated setting of federated learning, rather than distributing the clients across multiple devices. Therefore, we also use plaintext placeholders for the cryptographic steps that have identical output. Source code for our experiments will be made publicly available.

### 5.1 EXPERIMENTAL SETUP

**Datasets** We evaluate *FedProp* on three standard datasets for multi-class classification: CIFAR-10 (Krizhevsky, 2009), which has 10 classes and is used in previous federated SSL works, as well as the more difficult CIFAR-100 (Krizhevsky, 2009) and Mini-ImageNet (Vinyals et al., 2016) which both have 100 classes. To the best of our knowledge ours is the first work in this federated SSL setting to evaluate on these more challenging datasets. All three datasets consist of 60,000 images

which we split into training sets of size $N := 50,000$ and test sets of size $10,000$. From the training set, $N_L$ examples are labeled and the remaining $N - N_L$ are unlabeled. For CIFAR-10 we evaluate with $N_L = 1,000$ and $5,000$. For CIFAR-100 and Mini-ImageNet we take $N_L = 5,000$ and $10,000$.

**Federated Setup**    We simulate a FL scenario by splitting the training data (labeled and unlabeled) between $m$ clients. $m_L$ of these have partly labeled data, while the others have only unlabeled data. Each client is assigned a total of $N/m$ data points of which $N_L/m_L$ are labeled if the client is one of the $m_L$ which possess labels. We simulate statistical heterogeneity among the clients by controlling the number of classes each client has access to. In the i.i.d. setting all clients have uniform class distributions and receive an equal number of labels of each class. In the non-i.i.d. setting we assign a class distribution to each client and clients receive labels according to their own distribution.

**Networks**    Following prior work, we use 13-layer CNNs (Tarvainen & Valpola, 2017) for CIFAR-10 and 100 and a ResNet-18 (He et al., 2016) for Mini-ImageNet. Feature extractors are all layers except the last fully connected one, thus embeddings have dimension 128 and 512, respectively.

**Hyper-parameters**    We use `FederatedAveraging` as the `FederatedOptimization` scheme. We choose hyper-parameters for all methods based on training progress (LSH dimension, $k$-NN parameter) or accuracy on a held-out validation set consisting of 10% of the training data (batchsize, learning rate). Detailed values are provided in the appendix.

**Baselines**    We compare *FedProp* to a broad range of other methods, both from the existing federated SSL literature as well as our own baselines. For fairness all methods use the same network architectures, and hyper-parameters are chosen individually to maximize each method's performance. As representatives of consistency regularization we report results for three methods: *FedMatch* (Jeong et al., 2021), *FedSiam* (Long et al., 2020), and *FedAvg+MT*, where the last is our own adaptation of mean teacher (MT) (Tarvainen & Valpola, 2017) to the FL setting: In each training round the server broadcasts both a global (student) model and a teacher model to the clients. Clients locally train the student model and update the teacher model as the empirical moving average of the student. Clients then return both models to the server which separately averages the local students and teachers. To reflect pseudo-label (PL) based approaches, we include three methods: *FedAvg+PL(network)* repeatedly uses network predictions to pseudo-label sampled client data. *FedSem+*, which is based on *FedSem* (Albaseer et al., 2020), computes all pseudo-labels only once during training, but it additionally uses entropy-based sample weights, as we found this to consistently improve performance. *FedAvg+PL(localLP)*, based on *FedHar* (Presotto et al., 2022), uses per-client label propagation, otherwise it is identical to Algorithm 3. Additionally, we report results for training in a supervised manner on only the available labeled examples, *FedAvg (labeled only)*. This provides a lower bound for all SSL methods which have access to both the labeled and unlabeled data held by the clients.

**Combinations**    As a pseudo-label based approach, *FedProp* is orthogonal to techniques that utilize unlabeled data by modifying the loss function, such as the consistency regularization in *FedAvg+MT*. To demonstrate this, we also report results when combining both approaches as *FedProp +MT*.

## 5.2   EXPERIMENTAL RESULTS

We report the results of our experiments in Tables 1 and 2 as the average accuracy and standard deviation over three random splits of the data for each setting. Table 1 provides a comparison of *FedProp* to other approaches and baselines in the standard setting of CIFAR-10 with 100 clients, which has been previously used to evaluate FL and SSL methods. In each case, we report results when 1,000 or 5,000 of the data points are labeled. Either all or half of the clients have labels, which are either i.i.d. or non-i.i.d. across clients. In all cases but one, *FedProp* achieves the best results among all methods, in some cases by a large margin. The most competitive alternative is the *FedAvg+MT* baseline. As a method based on consistency-regularization, it can be readily combined with our pseudo-label based approach and we observe that this combination (*FedProp+MT*) achieves even better results across the board. Table 2 reports on the harder situation with many more classes, for which the previous methods have not been evaluated. Again, *FedProp* achieves better results than the baselines here, and for CIFAR-100 can be further improved by combining it *FedProp+MT*.

Besides a ranking of methods, Tables 1–2 also provide some insights: First, pseudo-labels based on LP tend to help more than those based on network outputs. Presumably, this is because they do not just reinforce information that is already in the classifier but potentially also correct prediction errors

Table 1: Classifical accuracy [in %] on CIFAR-10 (average and standard deviation across three runs)

| | CIFAR-10, i.i.d. ($m = 100$) | | | |
|---|---|---|---|---|
| | $m_L = 100$ | | $m_L = 50$ | |
| **Method** | $N_L = 1000$ | $N_L = 5000$ | $N_L = 1000$ | $N_L = 5000$ |
| FedAvg (labeled only) | $55.46 \pm 0.43$ | $76.13 \pm 0.46$ | $56.97 \pm 0.59$ | $80.36 \pm 0.07$ |
| FedAvg+PL(network) | $60.12 \pm 0.15$ | $79.45 \pm 0.31$ | $59.14 \pm 0.35$ | $81.04 \pm 0.20$ |
| FedAvg+PL(localLP) | $61.75 \pm 2.22$ | $85.11 \pm 0.73$ | $65.29 \pm 2.50$ | $84.41 \pm 0.25$ |
| FedMatch | $50.93 \pm 0.56$ | $72.22 \pm 0.14$ | $57.10 \pm 0.46$ | $77.80 \pm 0.32$ |
| FedSem+ | $59.98 \pm 0.49$ | $79.49 \pm 0.15$ | $59.67 \pm 0.47$ | $80.94 \pm 0.25$ |
| FedSiam | $67.02 \pm 0.98$ | $82.06 \pm 0.56$ | $62.98 \pm 1.61$ | $78.45 \pm 0.34$ |
| FedAvg+MT | $62.37 \pm 1.69$ | $84.92 \pm 0.64$ | $70.14 \pm 1.87$ | $85.34 \pm 0.18$ |
| FedProp (ours) | $70.91 \pm 0.71$ | $86.65 \pm 0.16$ | $70.81 \pm 1.65$ | $86.29 \pm 0.34$ |
| FedProp+MT (ours) | $\mathbf{72.58 \pm 0.36}$ | $\mathbf{88.17 \pm 0.18}$ | $\mathbf{73.63 \pm 1.99}$ | $\mathbf{87.54 \pm 0.14}$ |
| | CIFAR-10, non-i.i.d. ($m = 100$) | | | |
| | $m_L = 100$ | | $m_L = 50$ | |
| **Method** | $N_L = 1000$ | $N_L = 5000$ | $N_L = 1000$ | $N_L = 5000$ |
| FedAvg (labeled only) | $50.94 \pm 0.14$ | $75.34 \pm 1.38$ | $53.26 \pm 0.69$ | $79.65 \pm 0.12$ |
| FedAvg+PL(network) | $60.60 \pm 0.60$ | $80.07 \pm 0.53$ | $59.82 \pm 1.05$ | $81.14 \pm 0.23$ |
| FedAvg+PL(localLP) | $50.94 \pm 0.14$ | $76.61 \pm 1.50$ | $53.26 \pm 0.69$ | $79.65 \pm 0.12$ |
| FedMatch | $50.71 \pm 1.57$ | $71.99 \pm 0.70$ | $48.24 \pm 0.86$ | $66.37 \pm 0.41$ |
| FedSem+ | $60.93 \pm 0.97$ | $79.70 \pm 0.78$ | $59.74 \pm 0.74$ | $81.30 \pm 0.09$ |
| FedSiam | $67.85 \pm 0.26$ | $82.23 \pm 0.46$ | $62.29 \pm 1.84$ | $78.84 \pm 0.72$ |
| FedAvg+MT | $64.54 \pm 2.11$ | $84.07 \pm 0.60$ | $67.40 \pm 0.53$ | $85.58 \pm 0.35$ |
| FedProp (ours) | $\mathbf{73.76 \pm 0.71}$ | $85.53 \pm 0.56$ | $\mathbf{70.01 \pm 1.29}$ | $85.42 \pm 0.43$ |
| FedProp+MT (ours) | $70.56 \pm 0.75$ | $\mathbf{88.47 \pm 0.28}$ | $68.35 \pm 1.07$ | $\mathbf{87.71 \pm 0.49}$ |

Table 2: Classifical accuracy [in %] on CIFAR-100 and Mini-Imagenet (average and standard deviation across three runs)

| | CIFAR-100, i.i.d. | | Mini-Imagenet, i.i.d. | |
|---|---|---|---|---|
| | $m = m_L = 50$ | $m = m_L = 100$ | $m = m_L = 50$ | $m = m_L = 100$ |
| **Method** | $N_L = 5000$ | $N_L = 10000$ | $N_L = 5000$ | $N_L = 10000$ |
| FedAvg (labeled only) | $43.80 \pm 0.19$ | $53.91 \pm 0.25$ | $23.39 \pm 0.52$ | $31.72 \pm 0.54$ |
| FedAvg+PL(network) | $43.80 \pm 0.19$ | $54.19 \pm 0.21$ | $23.98 \pm 0.36$ | $31.86 \pm 0.57$ |
| FedAvg+PL(localLP) | $43.82 \pm 0.59$ | $54.38 \pm 0.36$ | $25.53 \pm 0.22$ | $33.09 \pm 0.62$ |
| FedAvg+MT | $49.09 \pm 0.38$ | $56.05 \pm 0.23$ | $25.98 \pm 0.70$ | $33.20 \pm 0.68$ |
| FedProp (ours) | $50.19 \pm 0.60$ | $57.00 \pm 0.08$ | $\mathbf{26.93 \pm 0.41}$ | $\mathbf{35.78 \pm 0.56}$ |
| FedProp+MT (ours) | $\mathbf{50.60 \pm 0.28}$ | $\mathbf{58.66 \pm 0.28}$ | $25.62 \pm 0.75$ | $33.24 \pm 0.71$ |

in regions close to the classifier decision boundaries. Second, the fact that *FedProp* always has a clear advantage over *FedAvg+PL(localLP)* supports the hypothesis that it is beneficial to obtain an estimate of the underlying data manifold by combining data from multiple clients. In the i.i.d. case this is particularly noticeable when the number of examples per class is small. In the non-i.i.d. case, where the local label distribution does not match the overall one, we even observed local LP to often reduce the accuracy, which caused the model selection procedure to completely deactivate it. This is in contrast to *FedProp*, which consistently improves the accuracy even in this case.

## 6 CONCLUSIONS

In this work we introduced the *FedProp* method for federated semi-supervised learning. Building on efficient cryptographic primitives, it allows the computation of label propagation along an estimate of the data manifold to which the data of all participating clients contribute, without the clients having to actually share their data with anyone else. Rather, the server received only a matrix of approximate similarity values between embedded data points. An interesting aspect for future work would be how to add additional security to this step, such as making it differentially private, which could allow an extension of the current approach to the malicious server setting. Our experiments established that cross-client label propagation can substantially increase the classification accuracy compared to training only on labeled examples, pseudo-labels based on network outputs or using per-client label propagation. In future work, it would be interesting to study not just *FedProp*'s accuracy, but also its efficiency on real-world federated learning tasks.

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

## A EXPERIMENTAL DETAILS

### A.1 HYPER-PARAMETER SETTINGS

Here we detail specific parameter settings in our experiments. For clarity we separate the parameters into those relating to federated learning, those relating to network training and those belonging to the `CrossClientLP` routine.

**Federated learning parameters** We set the number of clients to $m = 100$, except for our experiments on CIFAR-100 and Mini-Imagenet with $n_L = 5000$. In these cases we set $m = 50$ as it is not possible to create an i.i.d. split of the data over 100 clients since the number of classes (C=100) is too large. For CIFAR-10 we set the number of clients which possess labels to $m_L = 100$ and $m_L = 50$. On CIFAR-100 and Mini-Imagenet we set $m_L = m$.

For the `FederatedOptimization` method we choose *FederatedAveraging*. The `ClientUpdate` step therefore corresponds to $E$ epochs of stochastic gradient descent (SGD) of a loss function. We set the number local epochs to $E = 5$ and the loss function is (per sample weighted) cross-entropy loss. The `ServerUpdate` step corresponds to averaging the model updates:

$$\texttt{ServerUpdate}(\theta^{(j)} \text{ for } j \in P) = \frac{1}{|P|} \sum_{j \in P} \theta^{(j)}.$$

The number of training rounds is set to $T = 1500$ and the number of clients sampled by the server per training round is set to 5, so $\tau = 0.05$ when $m = 100$ and $\tau = 0.1$ when $m = 50$. Note that when $m_L < m$ we ensure that the server samples $\tau m_L$ clients from the labeled portion (and $\tau(m - m_L)$ from the unlabeled) to ensure that there are some labels present in the graph.

**Network training parameters** We use standard data augmentation following Tarvainen & Valpola (2017). On CIFAR-10 and CIFAR-100 this is performed by 4×4 random translations followed by a random horizontal flip. On Mini-ImageNet, each image is randomly rotated by 10 degrees before a random horizontal flip. We use weight decay for all network parameters which is set to $2 \times 10^{-4}$. When carrying out SGD in the `ClientUpdate` we use batches of data $B = B_L \cup B_U$ where $B_L$ is a batch of labeled data and $B_U$ is a batch of pseudo-labeled (previously unlabeled) data. We set $|B_L|$ according to how many labeled samples the client has available, $|B_L| = \min(50, \#labels)$. We set $|B_U| = |B_L|$. Learning rate for SGD is set according to this batch size. On CIFAR-10, for $|B_L| < 50$ we set the learning rate to 0.1 and for $|B_L| = 50$ we set the learning rate to 0.3. On CIFAR-100 and MiniImageNet we always have $|B_L| = 50$ and we set the learning rates to 0.5 and 1.0 respectively. We decay the learning rate using cosine annealing so that the learning rate would be 0 after 2000 rounds.

**CrossClientLP parameters** We set the LSH dimension to $L = 4096$ as this gave near exact approximation of the cosine similarities while still being computationally fast (less than 1 second per round). We set the sparsification parameter to $k = 10$, so that each point is connected to its 10 most similar neighbors in the graph, and the label propagation parameter to $\alpha = 0.99$.

