# OpenReview forum: "FedProp: Cross-client Label Propagation for Federated Semi-supervised Learning"
_ICLR.cc/2023/Conference — Submitted to ICLR 2023_

### Official Review · Reviewer_8bDJ · 2022-10-16

**Confidence:** 5
**Correctness:** 2
**Technical Novelty And Significance:** 2
**Empirical Novelty And Significance:** Not applicable
**Recommendation:** 3

**Clarity, Quality, Novelty And Reproducibility:**

The paper is easy to follow. The paper novelty is limited from my perspective.  The experimental setup is clear, hence I think the results can be reproducible.

**Strength And Weaknesses:**

Strengths
+ The studied problem is important
+ Paper is easy to follow

Weaknesses
- Novelty is limited.
- No theoretical results.
- Lack of  comparison with state-of-the-arts.
- Evaluation results are insufficient


**Summary Of The Paper:**

The paper studies federated semi-supervised learning and introduces the FedProp method. FedProp allows the computation of label propagation along an estimate of the data manifold to which the data of all participating clients contribute and use efficient cryptographic primitives to avoid accessing the data from different clients. Experiments on three standard benchmarks show that FedProp achieves promising semi-supervised classification performance. The main contribution lies in using LP to generate pseudo labels for unlabeled samples and applying cryptographic primitives to avoid direct data access



**Summary Of The Review:**

Detailed comments:

The main idea to leverage LP to generate pseudo labels is not new.  Also LP is shown to be not effectiveness enough to generate  pseudo labels, if the data distribution does not naturally form a manifold, e.g., when the data samples is limited.

Why connecting data form different clients to perform label propagation? In the non-IID settings, the data distribution across different clients could be  significantly different. Is there a need to connect data from different clients to perform LP? What if each client just use its data to perform LP and obtain pseudo labels?

Is the proposed FedRep convergent? In what convergence rate? What is the risk bound?

No formal privacy guarantees.   For instance, “first, the similarity is not computed between input data itself, but its feature representation according to the current model“. Using indirect feature representations cannot protect the input privacy. Particularly, many existing works has shown that using shared model gradients can reconstruct the data, so does the learnt representations.

Zhu et al. Deep leakage from gradients. In NIPS, 2018


There exist more advance SSL methods (e.g, mixmatch, FixMatch) which can augment data with pseudo labels. Applying mixmatch to the federated SSL setting should be not difficult? What is the federated SSL performance with these data augmentation performance?

SemiFL (NeurIPS 2021) is FedRGD are mentioned, but not compared. Actually, SemiFL has shown to significantly outperform the compared methods, e.g., FedMatch, especially in the non-IID setting.

What is the data distributed in CIFAR100 and Mini-Imagenet? IID or non-IID?

What is the impact of different number of labeled samples?

Why connecting data form different clients to perform label propagation, especially in the non-IID setting,  is unclear to me. I suggest the authors should conduct experiments to validate this.

There exist several federated learning methods for semi-supervised classification, where the client data itself is already a graph. Please discuss them.


Zhang et al.,  Subgraph Federated Learning with Missing Neighbor Generation. NeurIPS 2021.

Xie et al., Federated graph classifica- tion over non-iid graphs. NeurIPS 2021
Wang et al., Graphfl: A federated learning framework for semi-supervised node classification on graphs. ICDM 2022

---

> ### Author Response · Authors · 2022-11-10
> **Response to Reviewer 8bDJ (Part 1)**
>
> We kindly thank the reviewer for their feedback and are happy to provide clarification on all of the points raised.
>
> * “The main idea to leverage LP to generate pseudo labels is not new.”
>
> The main contribution of the paper is not the idea to leverage LP to generate pseudo labels and we did not claim it to be. Rather, as stated in the paper, the contribution (and novelty) lies in the CrossClientLP routine (section 4.1) which allows LP to be efficiently computed over the data of multiple clients without those clients having to share their data. Leveraging cross-client data interactions is beneficial for SSL (see below), and no other methods so far have been able to do so in the federated setting.
>
> * “Also LP is shown to be not effectiveness enough to generate pseudo labels, if the data distribution does not naturally form a manifold, e.g., when the data samples is limited.”
>
> Indeed, LP is most effective when the data allows estimating a manifold (graph). This is exactly why we 1) create the neighborhood graph in feature space rather than the original space, so the network can *learn* features that work well with LP, and 2) create the neighborhood graph from the data of multiple clients rather than individually per client, as this increases the amount of data used for this step.
>
> * “Why connecting data form different clients to perform label propagation? In the non-IID settings, the data distribution across different clients could be significantly different. Is there a need to connect data from different clients to perform LP?“
>
> We discuss this in the Introduction of our manuscript as well as Section 6.2. Combining data from multiple clients increases the amount of data available for estimating the data manifold and therefore improves the pseudo-label quality. Our experimental results confirm this. For non-iid data the beneficial effect might be smaller, but the worst that could happen is that no cross-client connections emerge, in which case our method reduces to per-client LP. Note, however, that our neighborhood graph is not on the original data but on the network features. These can be aligned between clients, even if their original data distribution was non-iid.
>
> * "What if each client just use its data to perform LP and obtain pseudo labels?”
>
> This is exactly the setting of our FedAvg+PL(localLP) baseline. That works clearly worse than the proposed cross-client method in all cases, including with non-IID data.
>
> * "Is the proposed FedRep convergent? In what convergence rate? What is the risk bound?"
>
> Empirically we observed FedProp to converge similarly to classic supervised Federated Averaging. Deriving specific theoretical bounds is an interesting question for future work.
>
> * “No formal privacy guarantees. For instance, “first, the similarity is not computed between input data itself, but its feature representation according to the current model“. Using indirect feature representations cannot protect the input privacy. Particularly, many existing works has shown that using shared model gradients can reconstruct the data, so does the learnt representations.”
>
> We do not claim that the feature representation provides formal privacy. Our privacy guarantees stem from the fact that the data (i.e. the feature vectors) never leave the client. All cross-client operations are performed in a cryptographically secure way. The only information the server receives is the set of approximate pairwise inner products (as we discuss in Section 4.2). The fact that these are computed from hashed network features is just an additional factor that certainly does not make the system *less* secure.
>
> * “There exist more advance SSL methods (e.g, mixmatch, FixMatch) which can augment data with pseudo labels. Applying mixmatch to the federated SSL setting should be not difficult? What is the federated SSL performance with these data augmentation performance?”
>
> This has been explored in prior work, see e.g. Table 1 of (Jeong et al. “FedMatch “, 2021) or Table 5 of (Long et al. “FedSiam”, 2020). They report that applying FixMatch in a federated SSL setting performs generally on par with or worse than FedMatch which our method significantly outperforms. We’ll be happy to include such results anyway in a revision of our manuscript.

---

> ### Author Response · Authors · 2022-11-10
> **Response to Reviewer 8bDJ (Part 2)**
>
> * “SemiFL (NeurIPS 2021) is FedRGD are mentioned, but not compared. Actually, SemiFL has shown to significantly outperform the compared methods, e.g., FedMatch, especially in the non-IID setting.”
>
> We do not compare our method to SemiFL or FedRGD because these methods solve a *different problem*. SemiFL and FedRGD assume that the server has access to labeled data and the clients have only unlabeled data. In contrast, we assume that the server has no labeled data at all and clients have partly labeled data. Labels-at-client and labels-at-server are both realistic but distinct federated SSL scenarios that require different approaches. SemiFL and FedRGD cannot be applied in the labels-at-client scenario that we study, and likewise FedProp cannot be applied to the labels-at-server scenario.
>
> * “What is the data distributed in CIFAR100 and Mini-Imagenet? IID or non-IID?”
>
> This is stated in the first row of Table 2: for CIFAR100 and Mini-Imagenet the data distribution is IID.
>
> * “What is the impact of different number of labeled samples?”
>
> Tables 1 and 2 show results for different numbers of labeled samples (1000 and 5000 for CIFAR10, 5000 and 10000 for CIFAR100 and Mini-imagenet). If this is not the information you are looking for, could you please clarify your question?
>
> * “Why connecting data form different clients to perform label propagation, especially in the non-IID setting, is unclear to me. I suggest the authors should conduct experiments to validate this.”
>
> Our manuscript already has such experiments: we show that cross-client LP significantly outperforms per-client LP and that FedProp outperforms other methods especially in the non-IID setting.
>
> * “There exist several federated learning methods for semi-supervised classification, where the client data itself is already a graph. Please discuss them.”
>
> Thank you for these references, which we’ll be happy to cite and discuss. Being given a graph structure rather than having to infer it from data is indeed a promising path for improvement. Note, however, that we aimed for FedProp to be widely applicable to the common setting where only data points are given, not graph structures.

---

### Official Review · Reviewer_jCXS · 2022-10-20

**Confidence:** 4
**Correctness:** 3
**Technical Novelty And Significance:** 2
**Empirical Novelty And Significance:** 2
**Recommendation:** 6

**Clarity, Quality, Novelty And Reproducibility:**

The paper is clear and reproducible within reasonable limits given the limited space

Novelty exists but is also limited. The method is novel for federated SSL, but is not novel in the broader and more interesting (in the sense that has a larger auditorium) of general SSL. Federated SSL is more like a niche from SSL and so is its auditorium

**Details Of Ethics Concerns:**

No concern

**Strength And Weaknesses:**

Pros:
  - The idea to use manifold for pseudo-labeling has not been used before in federated semi-supervised leaning, although it has been used in classical SSL.
- the paper has reported improvement in performance over similar federated SSL papers, but is still far from plain SSL performance

Cons:
 - the paper claims to used secure and cryptographic protocols for communication. The paper does not really explain why do we need those. I understand that there may be connection problems and communication to be disrupted and thus adding checksums and redundancy will help. I do not understand why using cryptographic protocols help ensuring intimacy of the data for each client since the framework do not assume external digital attacks. Furthermore, while the proposed method uses them, there is no evaluation of their efficiency. For instance how much damage to the communication can be absorbed without changes in performance? The fact that prior art used them, does not help claiming anything here.
- Evaluation is strong from one perspective but disappointing from another. It is carried on standard benchmarks which is very nice because it allows very good comparisons with previous similar works. It is weak because it does not propose a scenario where it really make sense to have protected data such as face related problems or where to assume that data being separated over clients, there is also a bias (either in data, either in labels) due to independent gathering and annotation. While this may not be fair to this paper, yet this is a major conference, and simply improving inside the niche, after a while becomes less interesting.

**Summary Of The Paper:**

The paper proposes a solution for federated semi-supervised learning. the proposal lies averaging the manifold in the decision space over multiple clients. The sharing is done using two cryptographical protocols. The paper evaluates on three datasets

**Summary Of The Review:**

The paper shows improvement over previous federated SSL methods. But I see this topic as a niche and this paper does too little to enlarge the auditorium. The innovation is limited to the topic.

---

> ### Author Response · Authors · 2022-11-10
> **Response to Reviewer jCXS (Part 1)**
>
> We kindly thank the reviewer for their feedback and are happy to provide clarification on a number of points raised.
>
> One of the key issues that the reviewer seems to have with the paper is that federated SSL is not an interesting problem because they believe it is a niche of SSL:
>
> * “The method is novel for federated SSL, but is not novel in the broader and more interesting (in the sense that has a larger auditorium) of general SSL. Federated SSL is more like a niche from SSL and so is its auditorium”
>
> We strongly disagree with this claim, which suggests that work on federated SSL would only be of interest to a subset of researchers working on SSL. However this is not the case as Federated SSL is a way of improving prediction performance of federated learning by exploiting unlabeled client data, which otherwise would remain unused. Therefore, it can be of interest to the whole FL research community (which is substantial these days), as well as practitioners in the field.
>
> Additionally, federated SSL is not simply an application of SSL (in the sense that solving solving SSL would solve federated SSL). Federated Learning (both supervised and semi-supervised) presents a number of challenges that are not found in a centralized setting e.g. maintaining privacy, communication efficiency, dealing with non-iid data, dealing with clients dropping out, etc. Moreover, specifically to federated SSL, past work (Jeong et al. FedMatch, ICLR 2021) has found that naively applying existing SSL methods (such as UDA or FixMatch) to the federated setting leads to poor performance. Therefore, improvements in centralized SSL do not necessarily translate to improvements in federated SSL. In order to make progress on leveraging unlabelled data in federated learning, research that is specific to federated SSL is needed. If papers are simply rejected on the basis that they do federated SSL then no progress will be made in the field.
>
> Regarding novelty, please see our answer on pseudo labels below.
>
> We shall now address the more specific concerns raised by the reviewer.
>
> * “the paper claims to used secure and cryptographic protocols for communication. The paper does not really explain why do we need those. I understand that there may be connection problems and communication to be disrupted and thus adding checksums and redundancy will help. I do not understand why using cryptographic protocols help ensuring intimacy of the data for each client since the framework do not assume external digital attacks.”
>
> We believe this to be a (serious) misunderstanding. We use cryptographic protocol not as protection against communication disruption or hacker attacks, but to ensure that clients do not have to share their data with other clients or the server. Note that this does not require any external attackers to make sense. Rather, client data might be intrinsically sensitive (e.g. search histories) and sharing it with other clients might even be forbidden by law (e.g. medical records). Learning without clients having to give away their data is fundamentally the core of federated learning.
>
> * “Furthermore, while the proposed method uses them, there is no evaluation of their efficiency. For instance how much damage to the communication can be absorbed without changes in performance? The fact that prior art used them, does not help claiming anything here.”
>
> See above, our concern is not communication disruption but privacy of cross-client operations. That aspect we discuss in our analysis in Section 4.2 in terms of computational cost, communication efficiency and robustness against clients dropping out.
>
> * “The idea to use manifold for pseudo-labeling has not been used before in federated semi-supervised leaning, although it has been used in classical SSL.”
>
> We do not claim that the use of label propagation over a manifold is novel or a contribution of our paper. Our primary contribution is that we enable multiple clients to cooperate when assigning pseudo-labels for their data without having to share their data with each other. No prior works have been able to do that. It requires carefully reformulating label propagation as a distributed problem (section 4.1) which clients and server jointly solve. This reformulation is both efficient and private (does not expose client data to other clients or the server). We then apply this in the context of federated SSL and obtain state-of-the-art results.
>
> * “the paper has reported improvement in performance over similar federated SSL papers, but is still far from plain SSL performance”
>
> As discussed earlier, the federated setting presents many challenges that are not present in the centralized setting. Hence it is unreasonable and unfair to expect performance to be on par with centralized methods where all data is centrally available. Our paper reports state-of-the-art results in the context of federated SSL.

---

> ### Author Response · Authors · 2022-11-10
> **Response to Reviewer jCXS (Part 2)**
>
> * “Evaluation is strong from one perspective but disappointing from another. It is carried on standard benchmarks which is very nice because it allows very good comparisons with previous similar works. It is weak because it does not propose a scenario where it really make sense to have protected data such as face related problems or where to assume that data being separated over clients, there is also a bias (either in data, either in labels) due to independent gathering and annotation.”
>
> As the reviewer notes it is important to use standard benchmarks in order to compare to prior works. Benchmarking on real-world private data is of great interest for practical applications, but contrary to the spirit of an academic conference, as the results would not be comparable to earlier work and would not be reproducible.
> At the same time, we do not agree with the reviewer that the use of standard datasets in this setting does not represent a realistic scenario. One of the major motivating applications of federated SSL is image recognition in the context of photos taken by cameras on smartphone devices. Such data is difficult to annotate, hence it is a realistic occurrence of SSL, and often sensitive hence ideally would remain private and stored on device. The datasets that we use, such as (mini)ImageNet, are reasonable proxies for such photo data.
> Finally, we would like to emphasize that our experiments do simulate the “bias due to independent gathering and annotation” in our experiments: all non-IID experiments in the paper are exactly simulating different data distributions across the clients.

---

> ### Comment · Reviewer_jCXS · 2022-11-21
> **post-rebuttal review**
>
> I have carefully read the rebuttal as well as other reviewers' opinions and the response to them.
> W.r.t my concern most of them have been alleviated. However, the paper still does not claim much : it is a solid federated SSL paper,  an improvement to previous federated SSL papers, but it does not go beyond that. Testing on another database (academic face recognition, face expression, medical image databases) would have been an opportunity, not a "must", but it was missed.
>
> In conclusion, I have improved  my rating, but given that federated SSL is rather a niche,  a strong paper here, but which does not go further, may not be a too god fit for ICLR

---

### Official Review · Reviewer_G4t3 · 2022-10-26

**Confidence:** 4
**Correctness:** 3
**Technical Novelty And Significance:** 2
**Empirical Novelty And Significance:** 2
**Recommendation:** 5

**Clarity, Quality, Novelty And Reproducibility:**

The paper is well written with minor linguistic errors. The structure is well organized and the content is easy to follow. There are some typos in the headers of Table 1 & 2.

The work lacks major novelty as it uses some available tools and techniques and combines them  to generate its solution. As the code is not shared it is difficult to reproduce the results



**Strength And Weaknesses:**

Strengths: Federated Label Propagation (FedProp) uses Label propagation (LP), Locality Sensitive Hashing (LSH), and some other established semi supervised tools/techniques and combines them in a structured fashion to guide semi supervised learning. The experiments and reported results look promising. Also, an analysis of the correctness, efficiency and robustness of their FedProp algorithm has been addressed in section 4.2.

Weakness: The results reported in section 5 don’t include any statistical tests so it's difficult to verify whether the results make any significant difference to existing techniques. The proposed methodology also shows some limitations in terms of scalability (as per the results in section 5): The reported gain in performance reduces when tested for a larger (CIFAR-100 vs CIFAR-10) or a more complex datasets (Mini-Imagement -100).

Another weakness/limitation is this work lacks major technical contributions; It uses existing tools and techniques and ties them as its solution.


**Summary Of The Paper:**

This paper presents  Federated Label Propagation (FedProp), an algorithm, to effectively learn semi-supervised models by distributing both labeled and unlabelled data across multiple clients.  Only model parameters are shared across clients through a server (following a client server architecture) but not the data, which is in fact one of the principles of federated learning. FedProp algorithm takes into account existing tools/techniques such as  Label propagation (LP) and Locality Sensitive Hashing (LSH) as part of its solution.

In summary, FedProp  first initializes the global solution by learning client-wise independent solutions from corresponding labeled data (that is assigned to a client) and finally running an aggregation step for parameters shared by clients. In addition to the aggregation, the server also acts as an intermediator to share parameters across clients through a broadcasting operation whenever necessary. At each iteration clients are responsible for independently assigning pseudo labels to its portion of unlabeled data and subsequently use those as new/additional labeled data to update its parameters before broadcasting to the server. The above process (pseudo labeling, parameter update and broadcasting and aggregation) iterates for a predefined number of iterations and is expected to converge (at the end).

The proposed technique has been tested on CIFAR-10, CIFAR-100 and Mini-Imagenet datasets and compared against existing techniques such as FedAvg, FeMatch, FedSem, and FedSiam. Reported results  are found to be comparable.


**Summary Of The Review:**

I have gone through the paper more than once including the appendices. Overall, the idea is quite simple: combine existing tools and techniques in a smart way that does the job.  Distribute labeled and unlabelled data across clients and let clients learn and share their parameters to a server; the server then performs some aggregation and shares them back to clients. Iterate these back and forth steps until the solution converges. The work lacks major technical contributions although the reported results look promising.

---

> ### Author Response · Authors · 2022-11-10
> **Response to Reviewer G4t3 (Part 1)**
>
> Thank you for the feedback. We are happy to provide clarification for a number of points raised.
>
> Based on the reviewers comments we believe there may have been some fundamental misunderstandings relating to both the problem setting and the main contribution of the paper.
>
> It appears the reviewer believes that data is somehow sent or assigned to the clients before training begins:
>
> * “to effectively learn semi-supervised models by distributing both labeled and unlabelled data across multiple clients”
> * “from corresponding labeled data (that is assigned to a client)”
> * “Overall, the idea is quite simple: combine existing tools and techniques in a smart way that does the job. Distribute labeled and unlabelled data across clients…”
>
> This is not the case. In Federated Learning the data is generated by the clients themselves (e.g. mobile phone users typing text on their keyboards). At no point has this data left the clients device and it is certainly not assigned to clients by the server (or anyone else) before training begins. We reiterate this as it is crucial for understanding the contribution of the method. There would be no need to maintain the privacy of data that already existed in some centralized form at the server before being distributed among the clients to enable training.
> Maybe this misunderstanding is due to the evaluation protocol commonly used in federated learning, in which a fixed dataset is split among multiple clients. That is really only for the sake of reproducibility, though, as experiments would not  be reproducible if they used real-world per-client private data.
>
> A second fundamental misunderstanding lies in the method itself. The reviewer appears to believe that clients independently compute pseudo-labels for their data (that is, without interacting with other clients):
>
> * “At each iteration clients are responsible for independently assigning pseudo labels to its portion of unlabeled data…”
>
> We wish to highlight in the strongest possible terms that this is *not* what the method proposes. The primary contribution of the paper is that we enable multiple clients to *cooperate* when assigning pseudo-labels for their data, which no prior works have yet been able to do. This requires carefully reformulating label propagation as a distributed problem (Section 4.1) which clients and server jointly solve. This reformulation is both efficient and private (does not expose client data to other clients or the server). Section 4.2 is an analysis of precisely this CrossClientLP step and not of FedProp as the reviewer incorrectly states.
>
> Given that the review does not mention CrossClientLP or discuss its working mechanisms in any detail in any section of the review, we are concerned that the reviewer might have overlooked this crucial component, which constitutes the main novelty of the paper. We hope that the reviewer will reassess his/her claim of a lack of novelty in this light.
>
> We now address the more specific points raised by the reviewer.
>
> * “The results reported in section 5 don’t include any statistical tests so it's difficult to verify whether the results make any significant difference to existing techniques.”
>
> Thank you for the suggestion. We have now performed t-tests to test the significance of the results. Firstly, we test significance for CIFAR10. Here, the tests show that the claim that FedProp outperforms the best performing method from the literature (FedSiam) is significant in all settings at the p < 0.005 level. Similarly the tests show that the claim that FedProp outperforms our own best performing baseline (FedAvg + MT) is significant in all but two settings at the p < 0.005 level, and in the remaining two settings at the p < 0.05 level. We also test significance for CIFAR100 and Mini-Imagenet. Here, the tests show that the claim that FedProp outperforms our own best performing baseline (FedAvg + MT) is significant in all but one setting at the p < 0.005 level, and in the remaining setting at the p < 0.06 level. We shall update the manuscript to include these results.
>
> * “Reported results are found to be comparable”
>
> As our results above show, our reported results are not just comparable but actually significantly better than the prior state of the art.

---

> ### Author Response · Authors · 2022-11-10
> **Response to Reviewer G4t3 (Part 2)**
>
> * “The proposed methodology also shows some limitations in terms of scalability (as per the results in section 5): The reported gain in performance reduces when tested for a larger (CIFAR-100 vs CIFAR-10) or a more complex datasets (Mini-Imagement -100).”
>
> Indeed, the absolute differences of our method to the best baseline (FedAvg+MT) is smaller for CIFAR-100 and Mini-ImageNet. However, that baseline is actually our own construction and not from prior work from the literature. With respect to prior results from the literature FedProp and FedProp+MT still provide substantial gains. The explanation for the gain compared to FedAvg+MT are smaller is presumably because the two datasets have many more classes (100 instead of 10). Therefore, much less training data per class is available and inferring reliable pseudo-labels becomes harder. Nevertheless, note that the gains in accuracy are still significant as stated above.
>
> * “As the code is not shared it is difficult to reproduce the results”
>
> We have now shared the code in a post directed at the reviewers and area chair.

---

### Official Review · Reviewer_ZBdp · 2022-11-23

**Confidence:** 3
**Correctness:** 3
**Technical Novelty And Significance:** 2
**Empirical Novelty And Significance:** 3
**Recommendation:** 3

**Clarity, Quality, Novelty And Reproducibility:**

-The paper is clearly written and easy to follow. There are some concerns regarding the novelty and feasibility, see above.

**Strength And Weaknesses:**

Pros

-The writing is clear and the paper is easy-to-follow. The authors attempt to solve a practically important question and show some favorable empirical results.

Cons

-There are several issues that remain unsolved in the paper. The first concern is about novelty. To utilize the semi-supervised training data, the proposed method simply (in terms of methodology) employs the widely used label propagation in federated training. Although in practice, in order not to violate the federated training protocols (do not share data between clients) by using cross-client data information (embeddings), the authors introduce some cryptographic tools which are claimed to be fully secure, but they do not provide any privacy guarantees.

-The second concern is about feasibility. Label propagation is not free. We may not accurately estimate the data manifold with only limited non-iid data and thus label propagation may not be reliable. The authors propose to solve this problem by doing so on embeddings of the training data from all clients. However, we still cannot verify the manifold assumption in this case and can only observe from the empirical performances. This raises another concern about the convergence of the proposed method (if the label propagation is not reliable). Existing experimental results are based on benchmark datasets and the semi-supervised data are artificially designed. It would be useful to test the proposed method with real-world non-iid semi-supervised client data and see if label propagation works well.


**Summary Of The Paper:**

-This paper studies the problem of federated learning with semi-supervised client (local) data. They propose a method that first pre-trains the client models with their labeled training data, then performs cross-client label propagation based on the embeddings of the labeled and unlabeled training data obtained from the pre-trained models, and after obtaining the pseudo labels and per-sample weights standard federated training is applied. They also provide some experimental results demonstrating the effectiveness of this method.

**Summary Of The Review:**

-From the reviewer's point of view, this paper provides some interesting empirical results but more theoretical investigations are needed.

---

> ### Author Response · Authors · 2022-12-05
> **Response to Reviewer ZBdp**
>
> Thank you for your feedback. We are happy to discuss the raised issues.
>
> Summary Of The Paper:
> * "This paper studies the problem of federated learning with semi-supervised client (local) data. They propose a method that first pre-trains the client models with their labeled training data, then performs cross-client label propagation based on the embeddings of the labeled and unlabeled training data obtained from the pre-trained models, and after obtaining the pseudo labels and per-sample weights standard federated training is applied. They also provide some experimental results demonstrating the effectiveness of this method."
>
> Indeed, we do those things, though really pseudo-labeling (PL) and federated training are interleaved, not performed sequentially. But the summary does not mention our main contribution: the CrossClientLP method which allows clients to jointly do PL in a federated learning (FL) setting without data leakage. As we highlight in the manuscript, this is the first FL method that allows exploiting cross-client interaction. In all prior FL work, the clients had to compute completely independently, so LP would have only been possible within each client’s local data, which usually is too little to reliably estimate a data manifold (and which we show experimentally does not work very well in practice).
>
> * "There are several issues that remain unsolved in the paper. The first concern is about novelty. To utilize the semi-supervised training data, the proposed method simply (in terms of methodology) employs the widely used label propagation in federated training. Although in practice, in order not to violate the federated training protocols (do not share data between clients) by using cross-client data information (embeddings), the authors introduce some cryptographic tools which are claimed to be fully secure, but they do not provide any privacy guarantees."
>
> We find this a misrepresentation of our work: as stated above (and repeatedly in the manuscript), our main contribution is the CrossClientLP method which allows clients to interact without having to share their data with each other or the central server. This requires carefully reformulating label propagation as a distributed problem (Section 4.1) which clients and server jointly solve. We analyze its correctness, privacy, efficiency and robustness in Section 4.2. The guarantees on data leakage are absolute, thanks to the cryptographic protocols (which are not just ‘claimed’ to be secure, but **proven so in the literature**). The guarantees on information leakage are formulated in a more subtle way, since -after all- some information has to be shared for the method to yield improvements. Please see the analysis of this in 4.2.
>
> * "The second concern is about feasibility. Label propagation is not free. We may not accurately estimate the data manifold with only limited non-iid data and thus label propagation may not be reliable."
>
> We are not sure what is meant by “not free”. Clearly, as every semi-supervised method, there are situations where PL helps and some where it does not, or even hurts. Theory on this is scarce, even in a non-federated setting. Therefore we show experimentally that in standard benchmark that were used in prior FL work, our method achieves consistently better results than just supervised training or alternative SSL approaches.
> But, to repeat, our contribution is not the idea of using label propagation, which has been for a long time and is a standard tool. Our contribution is a method that allows one to actually *use* PL in a federated setting without data leakage, which previously was not possible.
>
> * It would be useful to test the proposed method with real-world non-iid semi-supervised client data and see if label propagation works well.
>
> Please also see our replies to the other reviewers: for the submission, we concentrated on standard benchmarks in order to be able to show improvements over prior work. But we’ll be happy to include experiments on other datasets. Ideally, the reviewer(s) could suggest some. Also, our source code is available for everyone to try on their own data.

---

### Decision · Program_Chairs · 2023-01-20

**Decision:**

Reject

**Justification For Why Not Higher Score:**

Thanks for the detailed feedback to the reviewers, which clarified some of the concerns raised by the reviewers in their initial reviews. Overall, the paper discusses an interesting problem and the proposed solution is reasonable and effective. However, the technical contributions are limited since the proposed method combines known methods in semi-supervised learning and secure communication. The additional reviewer ZBdp also had the same opinion. Given these facts, I cannot recommend the acceptance of this paper.

**Justification For Why Not Lower Score:**

N/A

**Metareview: Summary, Strengths And Weaknesses:**

Summary:
The authors propose a semi-supervised federated learning method. Their method, called FedProg, adopts a manifold-based approach to semi-supervised learning and the data manifold is estimated jointly from the data of multiple clients and pseudo labels are computed through label propagation among clients. To keep the privacy of data, the authors proposed using multi-party Hamming distance computation and aggregation for secure communication. Experiments demonstrate the effectiveness of the proposed method.

Strength:
The problem setting is practical and the proposed method is demonstrated to be useful. The paper is nicely structured and written clearly.

Weakness:
Using manifold regularization for semi-supervised learning in federated learning is rather straightforward. Using secure communication is reasonable, but technical contributions are limited.